# Test-retest reliability of the Performance of Upper Limb (PUL) module for muscular dystrophy patients

Marta Gandolla[1]ᵒ*, Alberto Antonietti[1]ᵒ, Valeria Longatelli[1], Emilia Biffi[2], Eleonora Diella[2], Morena Delle Fave[2], Mauro Rossini[3], Franco Molteni[3], Grazia D'Angelo[2], Marco Bocciolone[4], Alessandra Pedrocchi[1]

**1** Nearlab, Department of Electronics, Information and Bioengineering, Politecnico di Milano, Milano, Italy, **2** Scientific Institute IRCCS E. Medea, Bosisio Parini, Italy, **3** Villa Beretta Rehabilitation Center, Valduce Hospital, Costa Masnaga, Italy, **4** Department of Mechanical Engineering, Politecnico di Milano, Milano, Italy

ᵒ These authors contributed equally to this work.
\* marta.gandolla@polimi.it

**Data Availability Statement:** All relevant data are within the paper and its Supporting Information files.

## Abstract

The Performance of the Upper Limb (PUL) module is an externally-assessed clinical scale, initially designed for the Duchenne muscular dystrophy population. It provides an upper extremity functional score suitable for both weaker ambulatory and non-ambulatory phases up to the severely impaired patients. It is capable of characterizing overall progression and severity of disease and of tracking the stereotypical proximal-to-distal progressive loss of upper limb function in muscular dystrophy. Since the PUL module has been validated only with Duchenne patients, its use also for Becker and Limb-Girdle muscular dystrophy patients has been here evaluated, to verify its reliability and extend its use. In particular, two different assessors performed this scale on 32 dystrophic subjects in two consecutive days. The results showed that the PUL module has high reliability, both absolute and relative, based on the calculation of Pearson's $r$ (0.9942), Intraclass Correlation Coefficient (0.9943), Standard Error of Measurement (1.36), Minimum Detectable Change (3.77), and Coefficient of Variation (3%). The Minimum Detectable Change, in particular, can be used in clinical trials to perform a comprehensive longitudinal evaluation of the effects of interventions with the lapse of time. According to this analysis, an intervention is effective if the difference in the PUL score between subsequent evaluation points is equal or higher than 4 points; otherwise, the observed effect is not relevant. Inter-rater reliability with ten different assessors was evaluated, and it has been demonstrated that deviation from the mean is lower than calculated Minimum Detectable Change. The present work provides evidence that the PUL module is a reliable and valid instrument for measuring upper limb ability in people with different forms of muscular dystrophy. Therefore, the PUL module might be extended to other pathologies and reliably used in multicenter settings.

**Funding:** This work has been performed thanks to the project USEFUL (Telethon GUP15021) and it was partially supported by Italian Ministry of Health (Ricerca Corrente VARA "2020" to Dr. G. Reni). The funders had no role in study design, data collection and analysis, decision to publish, or preparation of the manuscript. There was no additional external funding received for this study.

**Competing interests:** The authors have declared that no competing interests exist.

# Introduction

In the last years, increasing attention has been devoted by the scientific community to therapeutic and assistive approaches for people affected by neuromuscular degenerative diseases, such as muscular dystrophy (MD). Historically, clinical trials targeting neuromuscular degenerative diseases primarily included Duchenne muscular dystrophy (DMD) ambulant participants [1], in the effort to characterize the course of the disease. Classical and well-characterized outcome measures such as the 6-minutes walking test or the North Star Ambulatory Assessment have been used to characterize walking and lower extremities motor skills [2]. However, these scales do not provide any information about the clinical progression of DMD, particularly, as expected, for non-ambulant patients [2]. The increasing number of studies and technological development have highlighted the need to identify reliable outcome measures to assess upper limb functions [3], and to characterize them in terms of the minimum detectable change, enabling a robust comparison of single patient longitudinal sessions. Indeed, the lack of adequately validated clinical outcome measures for non-ambulatory MD patients has excluded them from participating in the therapeutic trials [4].

Upper limb residual abilities are clinically assessed with a variety of outcome measures across different pathologies, such as the Fugl-Meyer Motor Scale [5], the Action Research Arm Test (ARAT) [6], the Barthel Index, the Brooke scale [7, 8] or the Motor Function Measure (MFM) [9]. However, these scales are not able to assess proximal to distal muscular weakness progression, typical of MD diseases.

To overcome these limitations, Mayhew and colleagues developed a tool for upper limb measurements following the gradient of progression observed in MD patients, with particular attention to the DMD cohort, leading to the introduction of the Performance of the Upper Limb (PUL) module [1]. In particular, the PUL module, version 1.2, was specifically designed for the DMD population with a joint effort of clinicians, patients themselves, and caregivers. It provides a total upper extremity functional score able to characterize disease progression and severity and to track the stereotypical proximal-to-distal progressive loss of upper limb functions in neuromuscular pathologies.

The PUL module has been used in some trials with DMD participants [4, 10–14] to evaluate changes in motor performance of the upper limbs that occur over time, from walking child to the most compromised adults with only limited residual finger movements. It records the minimum variations in functionality, limiting ceiling and floor effects, through its scoring system [3], even if some ceiling effect has been observed in ambulant DMD boys [15].

Considering other forms of MD, such as Becker muscular dystrophy (BMD), Limb-Girdle muscular dystrophy (LGMD), or Facio-scapulo-humeral muscular dystrophy (FSHD), the Brooke scale and the MFM are unable to discriminate between different levels of severity in slowly progressive MD [16]. Given its higher granularity, the PUL module could be a useful evaluation tool to assess ability levels not only of DMD but also of other MD patients.

In the last years, the PUL module has been widely applied in DMD clinical trials [12, 17–20], but it has been rarely used in other dystrophies [21]. A previous study evaluated PUL 1.2 module in a group of 322 DMD boys and adults [10], showing that this scale is an excellent tool to assess both ambulant and non-ambulant DMD patients' upper limb functionality. However, test-retest validation has not been conducted on the participants' cohort. Moreover, the Minimum Detectable Change (MDC), a useful indicator to assess whether an observed improvement or worsening might be considered as relevant, was not characterized.

In this context, the objectives of this multicenter study are i) to determine PUL module reliability for multiple types of MD, both in terms of test-retest and of inter-rater reliability, ii) to characterize its minimum detectable change.

## Materials and methods

### Participants

Participants were recruited at the Scientific Institute IRCCS E. Medea (Bosisio Parini, LC, Italy) and at the Rehabilitation Institute Villa Beretta (Costa Masnaga, LC, Italy) from June 2017 to March 2019. A part of them was recruited from an ongoing Randomized Controlled Trial on the effectiveness of two commercial assistive devices for the upper limbs for dystrophic patients [22]; others were recruited from in-patients and out-patients services of the two clinical centers.

The inclusion criteria were as follows: i) diagnosis of DMD, BMD or LGMD (including all sub-types), ii) Medical Research Council Scale at deltoid and biceps brachii level equal or higher than 1, and iii) availability to sign the informed consent. No other inclusion criteria were established, therefore, patients with different levels of function of the upper limbs could be recruited. The level of ability at the time of recruitment was assessed through the Brooke scale [8]. The study has been approved by the Ethical Committee Boards of both clinical centers involved (Comitato Etico Interaziendale delle province di Lecco, Como e Sondrio, protocol ID: 130/2016; Comitato Etico dell'IRCCS E. Medea Sezione Scientifica dell'Associazione La Nostra Famiglia, protocol ID: 013/16), and written informed consent has been collected by all participants.

### Assessment procedure

The PUL module includes 22 items with an entry item to define the starting functional level (which corresponds to the Brooke scale) and 21 items subdivided into shoulder level (4 items), elbow level (9 items), and distal (i.e., wrist and fingers) level (8 items). For weaker patients, a low score on the entry item (i.e., less than 4 point of Item A) means that shoulder-level items do not need to be performed. Each dimension can be scored separately with a maximum score of 16 for the shoulder level, 34 for the elbow level, and 24 for the distal level. A total score can be achieved by adding the three-level scores, with a maximum global score of 74 points. We have assessed the PUL module reliability between two assessors considering the whole PUL scores for all MD types. Then, we conducted sub-group analyses considering the three MD types and the three PUL sections separately.

**Test-retest.**   PUL module was assessed in two consecutive days in one of the two clinical centers by two different trained assessors (two assessors for each clinical center involved). The tests were conducted on the dominant arm of each patient. As specified in the PUL module manual, the dominant arm has been defined according to the side used (or more frequently used) to draw or write. In any case, the participant him/herself was free to select the preferred side to use when performing unilateral PUL module sections, independently from the dominance. The timed items of the PUL were not included in the analysis. In fact, despite the fact that they are an important component of the test from a clinical point of view, especially when coming to longitudinal evaluation, they are not dependent on the assessor(s). However, the maximum available time was indeed respected. We labeled the two assessment sessions as PUL1 (i.e., test session on day 1), and PUL2 (i.e., retest session on day 2).

**Sub-group analyses.**   Two sub-groups analyses have been performed to better characterize the PUL module. In particular, the shoulder, elbow, and wrist/fingers sections have been investigated separately. Besides, we investigated the possible influence of the type of MD (i.e., DMD, BMD, LGMD).

**Inter-rater reliability.**   Inter-rater reliability for a higher number of assessors has also been investigated. As seen in the literature, we evaluated inter-rater reliability for ten different

assessors giving categorical ratings [23]. A further patient evaluation by means of the PUL module was filmed, and ten assessors were asked to evaluate the same patient with the PUL module by watching the video.

## Statistical analyses

**Power analysis.** A power analysis was conducted to determine the number of subjects required for this test-retest study [24]. The sample size calculation was derived from Walter and colleagues' study [24], considering a statistical power of 80% and a significance level ($\alpha$) of 0.05. The acceptable and expected reliability (i.e., $\rho_0$ and $\rho_1$) have been selected following Bujanga and Baharum guidelines [25]. In particular, $\rho_0$ was set equal to 0.50, representing the minimally acceptable reliability due by chance, and $\rho_1$ was set to 0.80.

**Systematic bias.** We performed the Shapiro-Wilk test to verify if PUL1 and PUL2 were normally distributed.

After the normality of populations was verified, we conducted a paired t-test to detect any systematic bias, which causes measurements to consistently either underestimate or overestimate the true value [26]. Indeed, there might be a trend for a retest to be higher than a prior test due to a learning effect.

**Reliability assessment.** We assessed the reliability of the PUL module with a set of statistical methods. Reliability can be defined as the consistency of measurements or as the absence of measurement errors [27].

i) *Pearson's r.* The correlation coefficient assesses relative reliability, which is the degree to which individuals maintain their position in a sample with repeated measures. To this aim, we computed the Pearson's r. Values in the range between 0.70 and 0.90 denoted high correlation, while values higher than 0.90 very high correlation [28].

ii) *Intraclass correlation coefficient (ICC).* Since the systematic bias is believed to be random and raters with similar characteristics were considered, the two-way random effect model was used and the *ICC*(2, 1) was selected. Values lower than 0.5, between 0.5 and 0.75, between 0.75 and 0.9, and higher than 0.90 indicate poor, moderate, good, and excellent reliability, respectively [29].

iii) *Standard Error of Measurement (SEM).* *SEM* is defined as the variation in patient-reported outcome scores attributed to instrument unreliability [30]. The more reliable the measurement response, the less error variability there would be around the mean. *SEM* was estimated as the square root of the mean square error term in the repeated measure ANOVA [27].

**Agreement between two raters assessment.** The agreement between PUL1 and PUL2 was investigated through the Bland-Altman (B-A) plot. Creating a B-A plot involves plotting the mean of two measurements against the difference between them. When inspecting the B-A plot, the 95% limits of agreement, which describe the range of differences between the two measurements (*mean*±2 *StandardDeviation*), are commonly represented. Points that fall within these limits of agreement indicate that PUL1 and PUL2 provide congruent results, while points outside the limits represent the cases of actual disagreement. Moreover, the mean difference between the measurements was computed to determine potential bias or lack of agreement. The bias can be considered significant if the *CIs* of the mean difference do not include the line of equality (i.e., zero) [31].

**Minimum Detectable Change.** The Minimum Detectable Change (*MDC*) is the smallest change in score that is likely to reflect a true change rather than a measurement error [30]. The *MDC* was calculated as $SEM * 1.96\sqrt{2}$. The *MDC* can be used to establish the minimum change in PUL module between two time points that can be considered clinically significant.

**Repeatability assessment.** The Coefficient of Variation (*CV*) is an often-quoted estimate of measurement error used to express the precision and repeatability of a measurement scale [32]. In order to indicate good repeatability, *CV* should be lower than 10% [27].

**Sub-group analyses.** For each of the two sub-group analyses (i.e., by PUL section and by MD type), we computed the same reliability indexes, as explained above: Pearson's *r*, *ICC*(2, 1), *SEM*, *MDC*, and *CV*.

**Inter-rater reliability.** We calculated the mean total PUL score obtained by the ten assessors, and the deviation from the mean for each assessor. PUL module has been considered to have sufficient inter-rater reliability if the deviation from the mean for each assessor was lower than the obtained *MDC* value.

## Results

### Participants

A sample size of 27 subjects was obtained by power analysis. Considering the availability of recruitable patients in the two centers involved, 32 patients were enrolled. 24 subjects were recruited at the Scientific Institute IRCCS E. Medea and 8 at the Rehabilitation Institute Villa Beretta. Patients' characteristics are listed in Table 1. Fifteen patients were affected by DMD, six by BMD, while eleven by LGMD. Among the LGMD group, four patients were diagnosed with LGMD2A, four patients with LGMD2E, one with LGMD2B, one with LGMD2D, and one with LGMD2L. The median age of the recruited participants was 22.5 years (IQR 26.5 years). Patients were characterized by a wide range of upper limb ability, as indicated by Brooke scores, ranging from 1 to 6, with a median value equal to 5.5 (IQR 1.91) points. 27 patients were right-handed, while 5 were left-handed.

PUL1 data ranged from 17 to 73 points, with a mean value of 42.13 (SD 17.25), while PUL2 data were included between 19 and 74 points, with mean value equals to 42.06 (SD 17.90). Fig 1 represents all the PUL scores as evaluated by the two assessors (PUL1 and PUL2). Outcome measures results are summarized in Table 2.

**Offset investigation.** The Shapiro-Wilk test assessed that PUL1 and PUL2 data came from two normally distributed populations (p-values$_{PUL1 = 0.0846}$, p-values$_{PUL2 = 0.1872}$). As a result, a parametric paired t-test was conducted between populations PUL1 and PUL2. A p-value of 0.9982 was obtained, showing that no systematic bias between the two populations was present.

**Reliability assessment.** A Pearson's correlation coefficient of *r* = 0.9942 between the two populations was obtained (p-value < 0.0001). Being Pearson's *r* higher than 0.90, it was concluded that the Test scores (i.e., PUL1) were highly correlated with the Retest scores (i.e., PUL2), indicating excellent reliability of the PUL module (Fig 2).

**Table 1. Patients' characteristics.**

|  | Duchenne MD | Becker MD | Limb-Girdle MD |
|---|---|---|---|
| Number of participants | 15 | 6 | 11 |
| Age (years), median [IQR] | 18 [14.5–22.5] | 43.5 [33.5–55.74] | 40 [21–50.5] |
| Brooke, median [IQR] | 5 [2–6] | 3 [2–5.5] | 6 [5–6] |
| Dominant arm, R/L | 13/2 | 5/1 | 9/2 |

MD = muscular dystrophy, R = Right, L = Left, IQR = Inter-Quartile Range.

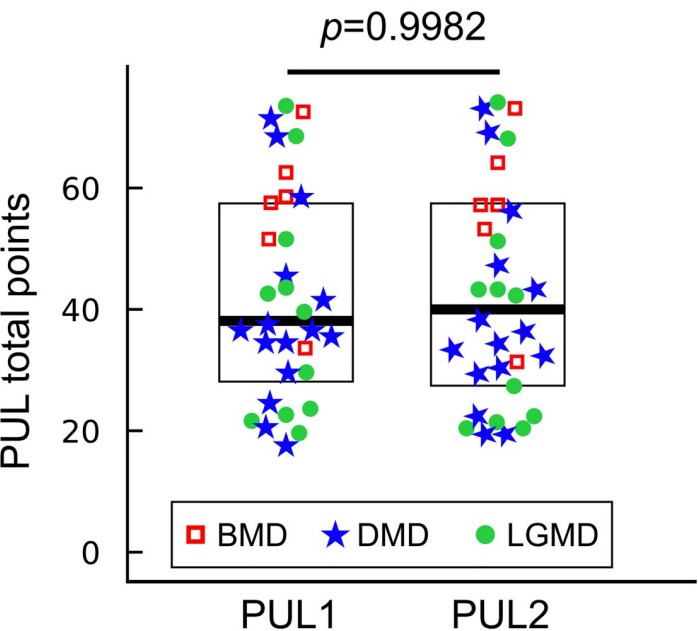

**Fig 1. PUL scores as evaluated by two assessors.** Each data point represents one PUL score for the test (PUL1) and the retest (PUL2) for $N = 32$ patients. Red squares ($N = 6$), blue stars ($N = 15$), and green dots ($N = 11$) are the PUL scores of BMD, DMD, and LGMD patients, respectively. Box-plots show median values of PUL1 and PUL2 (thick lines) and inter-quartile ranges. Paired t-test between PUL1 and PUL2 resulted in a $p - value = 0.9982$.

The $ICC(2, 1)$ resulted in being equal to 0.9943, representing excellent relative reliability. This coefficient can also be seen as the coefficient of inter-rater reliability, indicating the consistency of measurements and the extent to which the raters are interchangeable.

A *SEM* of 1.36 PUL-points was collected, which means that only approximately one point in the PUL module was due to the scale unreliability, representing a very good absolute reliability.

**Agreement between two raters assessment.** Inspection of the B-A plot (Fig 3) confirmed the absence of any systematic bias between PUL1 and PUL2. In fact, the B-A plot shows on the x-axis the mean score between the two assessors for each participant, and on the y-axis the difference between the two assessors in the evaluation of the same participant. Data were distributed above and below the mean difference, equals to 0.06. The 95% limits of agreement ranged

**Table 2. Results of the PUL module test-retest analysis.**

| Sample size | 32 |
| --- | --- |
| **Shapiro-Wilk test** | p-value ($PUL1$) = 0.0846 |
| | p-value ($PUL2$) = 0.1872 |
| **Paired t test** | p-value = 0.9982 |
| **Pearson's *r*** | 0.9942 |
| ***ICC*(2, 1)** | 0.9943 |
| ***SEM*** | 1.36 |
| ***MDC*** | 4 (3.77) |
| ***CV*** | 3% (SD 2.79) |

ICC = Intraclass Correlation Coefficient, SEM = Standard Error of Measurement, MDC = Minimum Detectable Change, CV = Coefficient of Variation.

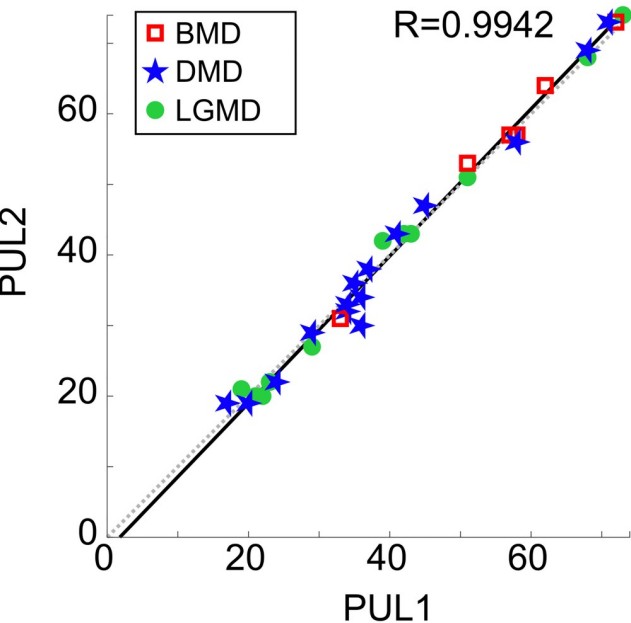

**Fig 2. Correlation of PUL scores from two assessors.** Each data point represents the PUL score of one patient (*N* = 32), on the x-axis the scores assigned during the test (PUL1), and on the y-axis the scores assigned during the retest (PUL2). Red squares (*N* = 6), blue stars (*N* = 15), and green dots (*N* = 11) are the PUL scores of BMD, DMD, and LGMD patients, respectively. The solid black line represents the linear regression fitting the PUL data points, while the dashed grey line represents the ideal correlation (*y* = *x*). Pearson's *r* value is 0.9942.

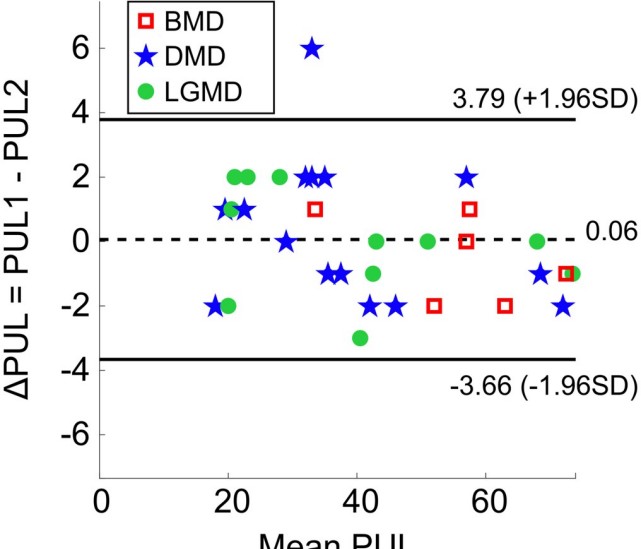

**Fig 3. Bland-Altman plot of PUL data obtained by the two assessors.** Each data point represents PUL scores of one patient (*N* = 32), on the x-axis the mean score ($\frac{PUL1+PUL2}{2}$) and on the y-axis the difference between test and retest score ($\Delta PUL = PUL1 - PUL2$). Red squares (*N* = 6), blue stars (*N* = 15), and green dots (*N* = 11) are the PUL scores of BMD, DMD, and LGMD patients, respectively. The dashed black line represents the mean difference value (0.06), while the solid black lines represent the 95% limits of agreement for each comparison (*mean* ± 1.96 *SD*).

**Table 3. Sub-group analysis of shoulder, elbow and wrist/fingers level.**

| PUL Level | Shoulder | Elbow | Wrist/Fingers |
|---|---|---|---|
| **Pearson's *r*** | 0.9896 | 0.9919 | 0.8955 |
| **ICC(2,1)** | 0.9861 | 0.9916 | 0.8980 |
| **SEM** | 0.64 | 1.07 | 0.73 |
| **MDC** | 2 (1.76) | 3 (2.97) | 2 (2.00) |
| **CV** | 6% (SD 9.10) | 7% (SD 10.71) | 2% (SD 3.07) |

*ICC* = Intraclass Correlation Coefficient, *SEM* = Standard Error of Measurement, *MDC* = Minimum Detectable Change, *CV* = Coefficient of Variation.

from -3.66 to +3.79 and all points, except for one, resulted in being included. Moreover, the 95% *CI* of the mean difference was -3.82 to 3.94, thus including the zero value. In this view, the agreement between PUL1 and PUL1 was verified.

 **Minimum detectable change.**  *MDC* resulted in being equal to 3.77 points, which was approximated to 4 points. Therefore, a change between two tests on the same subject can be considered significant only if equal or higher than 4 PUL-points; otherwise, the observed difference is not relevant.

 **Repeatability assessment.**  The coefficient of variation (i.e., $CV_i$) of each subject was computed, with a mean *CV* value of 3% (SD 2.79). The obtained results indicate good repeatability of the PUL module [27].

## Sub-group analyses

Sub-groups analysis investigating shoulder, elbow, and wrist/fingers sections separately confirmed the reliability of the PUL module (Table 3). The *CV* for the shoulder level was computed considering only those patients who were able to perform the shoulder items (n = 9). *MDC* has been calculated for each sub-group resulting in 2, 3, and 2 points for shoulder, elbow, and wrist/fingers sections, respectively Table 3. Sub-groups analysis investigating the influence of the type of MD (i.e., DMD, BMD, LGMD) is in line with the results obtained by the pooled population (Table 4), even if this analysis involved a reduced sample size.

## Inter-rater reliability

The mean total PUL score obtained by the ten assessors resulted in being 53.50 (SD 1.50) points. Maximum deviation from the mean resulted in being 2.00 points, and therefore lower than 4 points obtained as MDC.

**Table 4. Sub-group analysis on MD type.**

| Pathology | DMD (N = 15) | BMD (N = 6) | LGMD (N11) |
|---|---|---|---|
| **Pearson's *r*** | 0.9910 | 0.9956 | 0.9969 |
| **ICC(2,1)** | 0.9905 | 0.9929 | 0.9967 |
| **SEM** | 1.60 | 1.25 | 1.12 |
| **MDC** | 5 (4.43) | 4 (3.46) | 4 (3.10) |
| **CV** | 4% (SD 3.40) | 3% (SD 3.45) | 3% (SD 3.04) |

*ICC* = Intraclass Correlation Coefficient, *SEM* = Standard Error of Measurement, *MDC* = Minimum Detectable Change, *CV* = Coefficient of Variation.

## Discussion

Recent studies showed that clinical knowledge and technological developments have led to an increase in the life expectancy of MD patients. As a result, patients themselves identified the assistance of the upper limbs as a priority [33]. Existing scales designed for other neuromotor pathologies, such as the Brooke scale or the MFM, showed limitations for assessing MD patients' abilities and pathology progression. The Brooke scale provides an easy functional classification over six levels of function, but it is not sensitive to relatively minor functional changes, and it does not adequately follow MD disease progression for hand functions [34]. The MFM has floor and ceiling effects, and only a few items measure antigravity power in the forearm functions (i.e., item 15: forearms resting on a table, place both hands on top of the head, and item 23: arms along trunk, place forearms and hands on a table at the same time), which are clinically important for a significant number of MD patients in the years after the loss of ambulation. This scale was designed to assess functions in individuals with a broader range of neuromuscular disorders, such as spinal muscular atrophy or distal neuropathies with a different pattern of weakness [35].

The PUL module has been proposed as a more suitable tool to assess MD patients' upper limb functionality. The purpose of the PUL module is to evaluate changes in motor performance of the upper limbs that occur over time, from walking child to the most compromised adults with only limited residual finger movements, thus including a selection of items that cover a wide range of abilities, and recording minimal variations in functionality. PUL module has the advantage of providing evidence on the functional evolution over time of the disease from distal to proximal levels. After the beginning of this study, a novel version of the PUL module (PUL 2.0) has been validated and shared with the scientific community [13, 36, 37]. PUL 2.0 is constituted by 22 items, and some items of the previous version were deleted (i.e., items B, I, K, U, and V). The remaining items in the two versions measure the same construct, but the scoring system is different. Indeed, the PUL 2.0 has scoring options that vary across the scale between 0–1 to 0–2, according to performance. The PUL 1.2, instead, has a broader choice of scoring options (up to 0–5 in some items).

In DMD, the PUL module has been widely studied and used to evaluate changes in the muscular function of upper limbs along with time [36, 37]. However, in LGMD and BMD, only Artilhiero studied some patients to investigate the relationship between the PUL module and Jebsen–Taylor Test to assess and monitor upper limb function progression in patients with different MD types [21]. To the best of our knowledge, Manual Muscle testing, Brooke scale, and upper limb questionnaires are the more frequently used methods to test upper limbs in LGMD and FSHD [33].

In this study, we investigated PUL module reliability on people diagnosed with DMD, BMD, and LGMD, and we defined the minimum detectable change for longitudinal evaluations. In particular, two different assessors evaluated 32 subjects at different levels of residual ability, on two consecutive days. The results showed that the PUL module has high reliability, both absolute and relative, based on the calculation of Pearson's *r*, *ICC*, *SEM*, *MDC*, and *CV* and on the investigation of B-A plot. Pearson's *r* showed a very high correlation between measurements made by different assessors. B-A plot revealed agreement among the two measurements. The *ICC* of this test-retest was higher than 0.9, indicating excellent inter-rater reliability. The establishment of trustworthy test-retest reliability is essential for the clinical applications of this outcome evaluation tool. However, the *ICC* does not provide information about measurement errors. Therefore, the *SEM* should also be calculated, as high *ICC* does not necessarily mean a small measurement error [38]. A *SEM* of less than two points was obtained, representing high absolute reliability. The *MDC* is a compelling indicator that can be used in clinical trials to perform a comprehensive longitudinal evaluation of the effects of treatment

interventions with the lapse of time. According to this analysis, it can be concluded that an intervention is effective if the difference in PUL scores between subsequent evaluation points is equal or higher than 4 points; otherwise, the observed effect is not relevant. A *CV* of 3% was obtained, meaning adequate reliability of the scale [27]. It is worth a note that for DMD population only, *MDC* resulted to be equals to 5 points. This observation should be considered if only DMD patients are recruited, or for single-patient longitudinal evaluation.

The investigation of the reliability of the PUL module in its sub-scores (i.e., shoulder, elbow, and wrist/fingers) yielded similar results as the PUL module total score. Therefore, the PUL module could be involved to separately assess at proximal, middle, or distal level the clinical changes as the disease progresses or the effect induced by an intervention. In addition, given the MDC calculation at the three levels, single patient longitudinal changes might be assessed [23]. Furthermore, the PUL module has demonstrated to be a reliable evaluation tool considering different types of MD separately. It should be noted that patients diagnosed with disparate types of MD usually have different evolutions of their upper limb functionality, and also the same MD class shows inhomogeneities. For instance, different LGMD types may lead to different disease development, since LGMD2E behaves like DMD, while other types progress more slowly. Anyhow, this study deals with PUL module reliability, independently from the ability level. Finally, the PUL module demonstrated to have inter-rater reliability, as demonstrated by ten independent assessors. The comparability and the possibility of using PUL 1.2 and 2.0 versions in longitudinal studies have been shown [37]. This study demonstrated the robustness and reliability of the PUL module and, as a consequence, this clinical scale can be suggested as a primary outcome measure in experimental clinical trials involving patients diagnosed with muscular dystrophy other than DMD, such as LGMD and BMD.

## Conclusions

This study provides evidence that the PUL module is a reliable and valid instrument for measuring the upper limb ability of people with DMD, BMD, and LGMD. It is suitable to assess upper limb functionality, but also to provide evidence about intervention effect, by comparing scores obtained at different time points. Moreover, this novel study lays the groundwork for the inclusion of the PUL module in other muscular dystrophies, and not only in DMD. This information could be of help in clinical practice. Further studies collecting data in larger cohorts, inclusive of different neuromuscular or neurological diseases, will provide a better understanding of possible use of this scale as a standard outcome measure for clinical trials targeting upper limb functionality.

## Supporting information

**S1 Dataset.**
(XLSX)

## Acknowledgments

The authors thank the subjects for their participation in the study and all the assessors for their invaluable contribution to this study, in particular Dr. Mariska Janssen.

## Author Contributions

**Conceptualization:** Marta Gandolla, Alberto Antonietti, Emilia Biffi, Franco Molteni, Grazia D'Angelo, Alessandra Pedrocchi.

**Data curation:** Valeria Longatelli, Emilia Biffi, Eleonora Diella, Morena Delle Fave, Mauro Rossini.

**Formal analysis:** Alberto Antonietti, Valeria Longatelli.

**Funding acquisition:** Emilia Biffi, Franco Molteni, Grazia D'Angelo, Alessandra Pedrocchi.

**Methodology:** Marta Gandolla, Alberto Antonietti, Valeria Longatelli, Marco Bocciolone, Alessandra Pedrocchi.

**Supervision:** Marta Gandolla, Alessandra Pedrocchi.

**Writing – original draft:** Marta Gandolla, Alberto Antonietti, Valeria Longatelli.

**Writing – review & editing:** Marta Gandolla, Emilia Biffi, Eleonora Diella, Morena Delle Fave, Mauro Rossini, Franco Molteni, Grazia D'Angelo, Marco Bocciolone, Alessandra Pedrocchi.

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
