## [Decision Letter · Decision Letter 0]

27 Apr 2020

PONE-D-19-35028

Test-retest reliability of the Performance of Upper Limb (PUL) module for muscular dystrophy patients

PLOS ONE

Dear Dr. Gandolla,

Thank you for submitting your manuscript to PLOS ONE. After careful consideration, we feel that it has merit but does not fully meet PLOS ONE’s publication criteria as it currently stands. Therefore, we invite you to submit a revised version of the manuscript that addresses the points raised during the review process.

We would appreciate receiving your revised manuscript by Jun 11 2020 11:59PM. To enhance the reproducibility of your results, we recommend that if applicable you deposit your laboratory protocols in protocols.io, where a protocol can be assigned its own identifier (DOI) such that it can be cited independently in the future. For instructions see: http://journals.plos.org/plosone/s/submission-guidelines#loc-laboratory-protocols

We look forward to receiving your revised manuscript.

Kind regards,

Matti Douglas Allen, PhD

Academic Editor

PLOS ONE

Journal Requirements:

2. Thank you for including your ethics statement: The study has been approved by the Ethical Committee Boards of both clinical centers involved (Protocol IDs: 013/16; 130/2016), and written informed consent has been collected by all participants.

Reviewers' comments:

Reviewer's Responses to Questions

**Comments to the Author**

1. Is the manuscript technically sound, and do the data support the conclusions?

Reviewer #1: Yes

Reviewer #2: Partly

2. Has the statistical analysis been performed appropriately and rigorously? 

Reviewer #1: Yes

Reviewer #2: Yes

3. Have the authors made all data underlying the findings in their manuscript fully available?

Reviewer #1: Yes

Reviewer #2: Yes

4. Is the manuscript presented in an intelligible fashion and written in standard English?

Reviewer #1: Yes

Reviewer #2: No

5. Review Comments to the Author

Reviewer #1: Gandolla et al conducted a thorough test-retest study of the Performance of Upper Limb (PUL) module in a cohort of individuals with DMD, BMD, and LGMD. The PUL has been increasingly incorporated into studies and drug trials in DMD and may have utility in other types of muscular dystrophies. The rationale and experimental design of this study is solid; however, there are several items that need to be expanded upon or addressed to make it a valuable contribution to the body of literature for the PUL.

Methods section: Much attention was given to describing the statistical methods used, but details of the PUL test itself, its administration, and the cohort were missing.

• There is a newer version of the PUL called the PUL 2.0. Published articles on this version were not available at this beginning of this study, therefore use of the PUL 1.2 should not be considered a flaw in study design. However, the authors should acknowledge and reference the two different versions of the PUL and clearly state which version was used in this study. Recent articles citing the PUL 2.0 include "Functional levels and MRI patterns of muscle involvement in upper limbs in Duchenne muscular dystrophy. Mayhew et al 2019" and "Performance of Upper Limb module for Duchenne muscular dystrophy. Brogna et al 2018."

• How was dominant arm defined for the PUL?

• There was no mention of timed items of the PUL despite being an important component of the test.

• What types of LGMD were included in the cohort as presentation is quite heterogenous between different subtypes?

Results section:

• It would be interesting to have the data points in figures 2 and 3 labeled with the type of muscular dystrophy (ie different symbol shapes or colors for DMD, BMD, and LGMD).

• Although the study is probably not powered to do so, I would have liked to see reliability for BMD and LGMD in isolation as measures such as minimum detectable change could be different for different muscular dystrophy populations.

• The authors introduce the fact that the PUL has sub-scores for the high, mid, and distal level items, but they do not present any data on the reliability of sub-scores.

• No reference in the text to Table 2

• When reporting Brooke score in the results and table 1, the median score and range may be more appropriate than mean score and standard deviation.

Discussion section: The discussion section feels a bit underdeveloped. There is quite a bit of literature available in the DMD population regarding the PUL, and it would strengthen the manuscript to discuss how the findings of the present paper fit into the larger context of PUL literature.

Minor Edits:

• Do not capitalize muscular dystrophy. For example, it should be Duchenne muscular dystrophy and not Duchenne Muscular Dystrophy. Same for limb girdle and facioscapulohumeral.

• ARAT scale – spell out before abbreviating

• Page 3, line 76. Avoid the use of the phrase “They suffered from” and instead use “ They were diagnosed with DMD, BMD, or LGMD.”

• Recommend editing for English grammar/sentence structure as there are several instances of awkward sentences and minor grammar mistakes.

Reviewer #2: This work aims to provide evidence that the reliability of one of the new upper extremity ability assessment tools (PUL) that has been validated for the patients with Duchenne Muscular Dystrophy can be useful in a broader group of individuals with other muscular dystrophies. There is some benefit to accomplishing this aim, though the original work by Mayhew et al will likely and should continue to be the primary reference cited for validity of the PUL.

Major Concerns:

1) Throughout the manuscript, sentence structure and use of words (for example, the use of limb vs limbs) need to be improved. Also, there are many instances of writing in the negative. Some (but not all) examples of this include:

a. line 78 write “function” rather than “disability”

b. line 76 “suffered” should be “diagnosis”

c. line 259 “impairment” should be measuring “ability”

d. line 263, “has not been done up to today” instead state something like “we did this novel study”

2) The Introduction would be greatly improved by earlier on stating that reliability of the PUL for all MD has not been assessed. Because the PUL has been used for UE measurement for DMD for several years, it is not necessary to justify its use compared to other outcomes measures. As is, the reader does not learn that the paper is a reliability study for the four types of MD until too late in the paper. With this, there should be fairly equal introduction to LGMD and BMD as there is for DMD and no need to go into extraneous disease detail.

The first paragraph of the Discussion is really a great Intro and sufficient to replace several paragraphs in the current Introduction.

3) Inter-rater reliability reads as an addition later in the paper and should be detailed in the study design at the beginning of the methods (rather than later in lines 160-167 when it seems to become a new part of the protocol). It is not clear which data reflect the inter-rater reliability portion amongst the 10 assessors (or was it 4 assessors?).

4) Figure legends are missing making it difficult to fully assess the figures.

5) Does the Bland-Altman plot describe differences between the two assessments of one participant or does is describe differences between two raters? It is not clear as presented. In other words, on the y axis of Figure 3, what does the ‘two measures’ refer to (PUL1 vs PUL2….or assessor 1 vs assessor 2)?

6) The statistical section contain excessive detail on how standard analyses were conducted.

7) The Results and Discussion are underdeveloped. There is no interpretation of some of the results, for example, the Violin plots. What do those results mean?

8) Figure 2 would be enhances if the three different symbols would be used to indicate participants with DMD vs LGMD vs BMD.

6. PLOS authors have the option to publish the peer review history of their article (what does this mean?). If published, this will include your full peer review and any attached files.

Reviewer #1: No

Reviewer #2: No

---

## [Author Response · Author response to Decision Letter 0]

19 May 2020

Dear Editor and Reviewers,

Thank you for your careful analysis of our manuscript, and for allowing us to address the reviewers' comments.

We have greatly appreciated the comments and the suggestion provided. We are glad that the Reviewers have appreciated our study, and their comments show their expertise in the clinical evaluation of MD patients and the PUL scale that we are here validating for different types of MD.

We are uploading our point-by-point response to the comments (below), the manuscript with highlighted changes with respect to the previous version (in red), and the updated manuscript.

Reviewer #1:

Gandolla et al conducted a thorough test-retest study of the Performance of Upper Limb (PUL) module in a cohort of individuals with DMD, BMD, and LGMD. The PUL has been increasingly incorporated into studies and drug trials in DMD and may have utility in other types of muscular dystrophies. The rationale and experimental design of this study is solid; however, there are several items that need to be expanded upon or addressed to make it a valuable contribution to the body of literature for the PUL.

Methods section:

Much attention was given to describing the statistical methods used, but details of the PUL test itself, its administration, and the cohort were missing.

1) There is a newer version of the PUL called the PUL 2.0. Published articles on this version were not available at this beginning of this study, therefore use of the PUL 1.2 should not be considered a flaw in study design. However, the authors should acknowledge and reference the two different versions of the PUL and clearly state which version was used in this study. Recent articles citing the PUL 2.0 include "Functional levels and MRI patterns of muscle involvement in upper limbs in Duchenne muscular dystrophy. Mayhew et al 2019" and "Performance of Upper Limb module for Duchenne muscular dystrophy. Brogna et al 2018."

Thanks for the comment. As the reviewer already introduced, at the time when the study was designed and approved by the Ethical Committees, the PUL 2.0 was not yet validated. Indeed, we have explicitly included the version of the PUL module, which has been used, and we have commented on the PUL 2.0 module in the discussion sections as follows.

"After the beginning of the study, a novel version of the PUL module, and in particular PUL 2.0 module, has been validated and shared with the scientific community [Brogna et al. 2018, Mayhew 2020, Pane 2018]. PUL 2.0 is constituted by 22 items, and some items were deleted (i.e., item B, item I, item K, item U, and Item V). The remaining items in the two versions measure the same construct, but the scoring system is different. Indeed, the PUL 2.0 has scoring options that vary across the scale between 0–1 to 0–2, according to performance. The PUL 1.2, instead, has a broader choice of scoring options (up to 0–5 in some items)."

[Discussion]

Considering the involved cohort, in the Methods section, we better specified the inclusion criteria that we have followed. In the Results section, we characterized the different types of included LGMD2.

"The inclusion criteria were as follows: i) diagnosis of DMD, BMD or LGMD (including all sub-types), ii) Medical Research Council Scale at deltoid and biceps brachii level equal or higher than 1, and iii) availability to sign the informed consent."

[Methods]

"Among the LGMD group, four patients were diagnosed with LGMD2A, four patients with LGMD2E, one with LGMD2B, one with LGMD2D, and one with LGMD2L."

[Results]

2) How was dominant arm defined for the PUL?

The dominant arm for the execution of the PUL module was defined accordingly to the PUL module manual. In particular, the dominant arm is defined according to the side used (or more frequently used) to draw or write. In any case, the participant him/herself can select the preferred side to be used when performing unilateral PUL module sections, independently from the dominance. We have specified this in the methods section as follows.

"As specified in the PUL module manual, the dominant arm has been defined according to the side used (or more frequently used) to draw or write. In any case, the participant him/herself can select the preferred side to be used when performing unilateral PUL module sections, independently from the dominance."

[Materials and methods - Participants and assessment procedure]

3) There was no mention of timed items of the PUL despite being an important component of the test.

As noted by the reviewer, the timed items of the PUL have not been included in the analysis. This is because, although they are an essential component of the test from a clinical point of view, especially when coming to longitudinal evaluation, they are not dependent on the assessor(s). However, the maximum available time has indeed been respected. We have specified this as follows.

"The timed items of the PUL were not included in the analysis. In fact, despite the fact that they are an important component of the test from a clinical point of view, especially when coming to longitudinal evaluation, they are not dependent from the assessor(s). However, the maximum available time has been indeed respected."

[Materials and methods - Participants and assessment procedure]

4) What types of LGMD were included in the cohort as presentation is quite heterogenous between different subtypes?

As introduced at point 1, considering LGMD, all types were considered for possible inclusion in the study. In particular, the included participants were diagnosed with LGMD2A, LGMD2E, LGMD2B or LGMD2L. We included specific sub-types of LGMD in the appropriate section as follows.

 "Fifteen patients were affected by DMD, six by BMD, while eleven by LGMD. Among the LGMD group, four patients were diagnosed with LGMD2A, four patients with LGMD2E, one with LGMD2B, one with LGMD2D, and one with LGMD2L."

[Results – Participants]

Results section:

5) It would be interesting to have the data points in figures 2 and 3 labeled with the type of muscular dystrophy (ie different symbol shapes or colors for DMD, BMD, and LGMD).

Thanks for the comment. We have included labels for the different types of muscular dystrophy in all the figures (i.e., see Figures 1-3).

6) Although the study is probably not powered to do so, I would have liked to see reliability for BMD and LGMD in isolation as measures such as minimum detectable change could be different for different muscular dystrophy populations.

7) The authors introduce the fact that the PUL has sub-scores for the high, mid, and distal level items, but they do not present any data on the reliability of sub-scores.

As pointed out by the reviewer, the study sample size has not been calculated to allow for sub-groups analysis. However, since it is of interest, we have included a section both in the methods and in the results to investigate PUL module reliability for different muscular dystrophy types. In addition, as suggested by the reviewer, we have included the investigation of the PUL module reliability of the high, mid, and distal level sub-scores in methods and results sections.

“Two sub-groups analyses have been performed to better characterize the PUL module. In particular, the shoulder, elbow, and wrist/fingers sections have been investigated separately. Besides, we investigated the possible influence of the type of MD (i.e., DMD, BMD, LGMD).”

[Materials and methods – Assessment procedure - Sub-groups analyses.]

“For each of the two sub-group analyses (i.e., by MD type and by PUL section), we computed the same reliability indexes, as explained above: Pearson's r, ICC(2, 1), SEM, MDC, and CV.

[Materials and methods – Statistical analysis - Sub-groups analyses.]

“Sub-groups analysis investigating shoulder, elbow, and wrist/fingers sections separately confirmed the reliability of the PUL module (Table 3). The CV for the shoulder level was not computed, given that most of the patients (N=23) did not score enough in the entry item to perform the shoulder items. MDC has been calculated for each sub-group resulting in 2, 3, and 2 points for shoulder, elbow, and wrist/fingers sections, respectively (Table 3). Sub-groups analysis investigating the influence of the type of MD (i.e., DMD, BMD, LGMD) is in line with the results obtained by the pooled population (Table 4), even if this analysis involved a reduced sample size.”

[Results – Sub-groups analyses.]

8) No reference in the text to Table 2

We thank the reviewer for pointing it out. We have added the reference to this table in the appropriate section.

9) When reporting Brooke score in the results and table 1, the median score and range may be more appropriate than mean score and standard deviation.

As suggested by the reviewer, we reported the Brooke scores results in the form of median and range.

Discussion section:

10) The discussion section feels a bit underdeveloped. There is quite a bit of literature available in the DMD population regarding the PUL, and it would strengthen the manuscript to discuss how the findings of the present paper fit into the larger context of PUL literature.

Following reviewers' suggestions, we have re-written the discussion section. In particular, we have discussed PUL module advantages with respect to other clinical scales, the comparison with the PUL 2.0 module, the sub-groups analyses, and broader comments about the clinical impact.

Minor Edits:

• Do not capitalize muscular dystrophy. For example, it should be Duchenne muscular dystrophy and not Duchenne Muscular Dystrophy. Same for limb girdle and facioscapulohumeral.

Thanks, updated.

• ARAT scale – spell out before abbreviating

Thanks, updated.

• Page 3, line 76. Avoid the use of the phrase "They suffered from" and instead use "They were diagnosed with DMD, BMD, or LGMD."

Thanks, updated.

• Recommend editing for English grammar/sentence structure as there are several instances of awkward sentences and minor grammar mistakes.

To address the reviewer's comment, we thoroughly read the manuscript multiple times, and we checked it for grammatical mistakes, improving the structure of sentences. We thank the reviewer for the editing suggestions above, which we have followed in the revision of the manuscript.

Reviewer #2:

This work aims to provide evidence that the reliability of one of the new upper extremity ability assessment tools (PUL) that has been validated for the patients with Duchenne Muscular Dystrophy can be useful in a broader group of individuals with other muscular dystrophies. There is some benefit to accomplishing this aim, though the original work by Mayhew et al will likely and should continue to be the primary reference cited for validity of the PUL.

Major Concerns:

1) Throughout the manuscript, sentence structure and use of words (for example, the use of limb vs limbs) need to be improved. Also, there are many instances of writing in the negative. Some (but not all) examples of this include:

a. line 78 write "function" rather than "disability"

b. line 76 "suffered" should be "diagnosis"

c. line 259 "impairment" should be measuring "ability"

d. line 263, "has not been done up to today" instead state something like "we did this novel study".

To address the reviewer's comment, we thoroughly read the manuscript multiple times, and we checked it for grammatical mistakes, improving the sentence structure. We thank the reviewer for the editing suggestions, which we have followed in the revision of the manuscript.

2) The Introduction would be greatly improved by earlier on stating that reliability of the PUL for all MD has not been assessed. Because the PUL has been used for UE measurement for DMD for several years, it is not necessary to justify its use compared to other outcomes measures. As is, the reader does not learn that the paper is a reliability study for the four types of MD until too late in the paper. With this, there should be fairly equal introduction to LGMD and BMD as there is for DMD and no need to go into extraneous disease detail.

The first paragraph of the Discussion is really a great Intro and sufficient to replace several paragraphs in the current Introduction

We thank the reviewer for the suggestion, we have shortened and restructured the whole Introduction section, anticipating the main aims of the study and avoiding unnecessary clinical details, giving an even balance to the different MD types.

3) Inter-rater reliability reads as an addition later in the paper and should be detailed in the study design at the beginning of the methods (rather than later in lines 160-167 when it seems to become a new part of the protocol). It is not clear which data reflect the inter-rater reliability portion amongst the 10 assessors (or was it 4 assessors?).

Thanks for pointing this out, we have moved the inter-rater reliability section, that is now a paragraph of the section "Assessment procedure". The statistical analyses used for the inter-rater reliability have been moved under the "Statistical analysis" section.

We have also rephrased the "Test-retest" subsection, in order to clarify who were the assessors: 2 couples of assessors for the Test-retest analysis, and ten assessors for the inter-raters reliability.

Test-retest

PUL module was assessed in two consecutive days in one of the two clinical centers by two different trained assessors (two assessors for each clinical center involved).

[…]

Inter-rater reliability

Inter-rater reliability for a higher number of assessors has also been investigated. As seen in the literature, we evaluated inter-rater reliability for ten different assessors giving categorical ratings (Gandolla et al., 2015). A further patient evaluation by means of the PUL module was filmed, and ten assessors were asked to evaluate the same patient with the PUL module by watching the video.

[Materials and Methods]

4) Figure legends are missing making it difficult to fully assess the figures.

Thank you for the suggestion. According to both reviewers' comments, we have improved both the figures, adding informative legends and expanding the caption descriptions with greater detail. To help with figures immediate interpretation, we also provide figures with embedded captions.

5) Does the Bland-Altman plot describe differences between the two assessments of one participant or does is describe differences between two raters? It is not clear as presented. In other words, on the y axis of Figure 3, what does the 'two measures' refer to (PUL1 vs PUL2….or assessor 1 vs assessor 2)?

We have included in the relevant section a more specific explanation of the reported B-A plot, we have also added a more precise description of the figure in Fig.3 caption.

"Inspection of B-A plot (Fig. 3) confirmed the absence of any systematic bias between PUL1 and PUL2. In fact, B-A plot shows on the x-axis the mean score between the two assessors for each participant, and on the y-axis the difference between the two assessors in the evaluation of the same participant."

[Results - Agreement between two raters assessment.]

6) The statistical section contain excessive detail on how standard analyses were conducted.

As suggested by the reviewer, we summarized the Statistical analyses section, removing excessive details.

7) The Results and Discussion are underdeveloped. There is no interpretation of some of the results, for example, the Violin plots. What do those results mean?

As for the results section, we have now removed the Violin plot, and we have included a box-plot figure to represent the data, which in the authors' opinion better represents the population investigated (Fig. 1). Besides, two sub-groups analyses have been performed to better characterize the PUL module. In particular, the shoulder, elbow, and wrist/fingers sections have been investigated separately. Furthermore, the (possible) influence of the type of MD (i.e., DMD, BMD, LGMD) has been investigated.

Following reviewers' suggestions, we have re-written the discussion section. In particular, we have discussed PUL module advantages with respect to other clinical scales, the comparison with the PUL 2.0 module, the sub-groups analyses, and broader comments about the clinical impact.

8) Figure 2 would be enhances if the three different symbols would be used to indicate participants with DMD vs LGMD vs BMD.

Following the reviewer's suggestion, we modified the figure using different symbols to indicate different types of Muscular Dystrophy. In particular, we indicated DMD with blue stars, BMD with red squares, and LGMD2 with green dots.

---

## [Decision Letter · Decision Letter 1]

20 Jul 2020

PONE-D-19-35028R1

Test-retest reliability of the Performance of Upper Limb (PUL) module for muscular dystrophy patients

PLOS ONE

Dear Dr. Gandolla,

Thank you for submitting your manuscript to PLOS ONE. After careful consideration, we feel that it has merit but does not fully meet PLOS ONE’s publication criteria as it currently stands. Therefore, we invite you to submit a revised version of the manuscript that addresses the points raised during the review process.

If the relatively minor concerns raised by the reviewer are addressed, the manuscript should be ready for acceptance for publication.

We look forward to receiving your revised manuscript.

Kind regards,

Matti Douglas Allen, PhD

Academic Editor

PLOS ONE

Reviewers' comments:

Reviewer's Responses to Questions

**Comments to the Author**

1. If the authors have adequately addressed your comments raised in a previous round of review and you feel that this manuscript is now acceptable for publication, you may indicate that here to bypass the “Comments to the Author” section, enter your conflict of interest statement in the “Confidential to Editor” section, and submit your "Accept" recommendation.

Reviewer #1: (No Response)

Reviewer #2: All comments have been addressed

2. Is the manuscript technically sound, and do the data support the conclusions?

Reviewer #1: Yes

Reviewer #2: Yes

3. Has the statistical analysis been performed appropriately and rigorously? 

Reviewer #1: Yes

Reviewer #2: Yes

4. Have the authors made all data underlying the findings in their manuscript fully available?

Reviewer #1: Yes

Reviewer #2: Yes

5. Is the manuscript presented in an intelligible fashion and written in standard English?

Reviewer #1: Yes

Reviewer #2: Yes

6. Review Comments to the Author

Reviewer #1: I appreciate the authors addressing all of the reviewers’ previous comments. The changes have improved the manuscript clarity and detail. Just a few minor comments that would benefit from being addressed are included below. All other recommendations were addressed. No need for further peer review if the suggestions below are addressed.

• Minor grammatical errors remain throughout, though most have been corrected. For example, the final sentence of the abstract is a run-on sentence.

• In the introduction and discussion, the authors state that the PUL can evaluate ambulatory individuals without floor or ceiling effects. I feel this statement is too strong as Pane et al 2014 demonstrated that quite a few ambulatory boys had ceiling effects on the PUL, especially if they were still strong ambulators. (“The 6 Minute Walk Test and Performance of Upper Limb in Ambulant DMD Boys”)

• It seems that perhaps the data in figure 1 and 2 from PUL1 and PUL 2 are swapped (or they are reported backwards in the text). In the text it says the min score from PUL1 was 17 and min from PUL2 was 19, but in Fig 1, the lowest symbol is definitely lower in PUL2. I think this is the same problem as in Fig 2.

• Would be nice to have a SD for the CVs. Also, there was still an n=9 for CV for the shoulder level subgroup analysis, and you report a CV for the BMD subgroup analysis with an n=6. I feel the CV for the shoulder dimension should be reported as the reason given for not reporting it is not valid.

• It would be nice to point out in the discussion that the MDC in DMD was 5 rather than 4 as it was for the entire cohort and BMD/LGMD.

• The following sentence in the conclusion should be toned down a bit: “Thanks to this novel study, the use of the PUL module can be extended to clinical trials involving BMD and LGMD, and not only DMD.” I think this study lays the groundwork for the inclusion of the PUL in other muscular dystrophies, but as you mention, further validation is definitely required.

Reviewer #2: My previous concerns have been thoughtfully and thoroughly addressed. The paper will be a value to the field.

7. PLOS authors have the option to publish the peer review history of their article (what does this mean?). If published, this will include your full peer review and any attached files.

Reviewer #1: No

Reviewer #2: No

---

## [Author Response · Author response to Decision Letter 1]

23 Jul 2020

Dear Editor and Reviewers,

Thank you for your careful analysis of our manuscript, and for allowing us to address the reviewers' minor comments.

We have greatly appreciated the comments and the suggestion provided. We are glad that the Reviewers have appreciated our study..

We are uploading our point-by-point response to the comments (below), the manuscript with highlighted changes with respect to the previous version (in red), and the updated manuscript.

Reviewer #1:

Reviewer #1: I appreciate the authors addressing all of the reviewers’ previous comments. The changes have improved the manuscript clarity and detail. Just a few minor comments that would benefit from being addressed are included below. All other recommendations were addressed. No need for further peer review if the suggestions below are addressed.

• Minor grammatical errors remain throughout, though most have been corrected. For example, the final sentence of the abstract is a run-on sentence.

We have revised language, and we corrected the final sentence of the abstract. In addition, we have sent the manuscript to a mother tongue colleague in USA, who performed a thorough English grammar review.

• In the introduction and discussion, the authors state that the PUL can evaluate ambulatory individuals without floor or ceiling effects. I feel this statement is too strong as Pane et al 2014 demonstrated that quite a few ambulatory boys had ceiling effects on the PUL, especially if they were still strong ambulators. (“The 6 Minute Walk Test and Performance of Upper Limb in Ambulant DMD Boys”).

We have updated the sentence in the Introduction section as follows:

“It records the minimum variations in functionality, limiting ceiling and floor effects, through its scoring system (Mercuri et al., 2011), even if some ceiling effect has been observed in ambulant DMD boys (Pane et al., 2014).”

Instead, we removed the comment in the Discussion section, since it was already presented in the Introduction section.

• It seems that perhaps the data in figure 1 and 2 from PUL1 and PUL 2 are swapped (or they are reported backwards in the text). In the text it says the min score from PUL1 was 17 and min from PUL2 was 19, but in Fig 1, the lowest symbol is definitely lower in PUL2. I think this is the same problem as in Fig 2.

We thank the reviewer for spotting this out. We updated both figure 1 and 2.

• Would be nice to have a SD for the CVs. Also, there was still an n=9 for CV for the shoulder level subgroup analysis, and you report a CV for the BMD subgroup analysis with an n=6. I feel the CV for the shoulder dimension should be reported as the reason given for not reporting it is not valid.

As suggested, we added the SD for the CV for the entire population, as well as for the subgroup analyses. Moreover, we added the computation of the CV for the shoulder level, specifying that it is related to a limited number of patients - those who were able to perform the related items.

• It would be nice to point out in the discussion that the MDC in DMD was 5 rather than 4 as it was for the entire cohort and BMD/LGMD.

Thanks for highlighting this observation. We have included the following sentence in the Discussion section:

“It is worth a note that for DMD population only, MDC resulted to be equals to 5 points. This observation should be considered if only DMD patients are recruited, or for single-patient longitudinal evaluation.”

• The following sentence in the conclusion should be toned down a bit: “Thanks to this novel study, the use of the PUL module can be extended to clinical trials involving BMD and LGMD, and not only DMD.” I think this study lays the groundwork for the inclusion of the PUL in other muscular dystrophies, but as you mention, further validation is definitely required.

As suggested, we modified the sentence in the conclusion section as follows:

“Moreover, this novel study lays the groundwork for the inclusion of the PUL module in other muscular dystrophies, and not only in DMD.”

Reviewer #2:

My previous concerns have been thoughtfully and thoroughly addressed. The paper will be a value to the field.

We thank the reviewer for appreciating our work, and for the contribution with previsous revisions.

---

## [Editor Report · Decision Letter 2]

18 Sep 2020

Test-retest reliability of the Performance of Upper Limb (PUL) module for muscular dystrophy patients

PONE-D-19-35028R2

Dear Dr. Gandolla,

We’re pleased to inform you that your manuscript has been judged scientifically suitable for publication and will be formally accepted for publication once it meets all outstanding technical requirements.

Kind regards,

Matti Douglas Allen, PhD

Academic Editor

PLOS ONE
---

## [Editor Report · Acceptance letter]

18 Sep 2020

PONE-D-19-35028R2

Test-retest reliability of the Performance of Upper Limb (PUL) module for muscular dystrophy patients

Dear Dr. Gandolla:

I'm pleased to inform you that your manuscript has been deemed suitable for publication in PLOS ONE. Congratulations! Your manuscript is now with our production department.

Kind regards,

on behalf of

Dr. Matti Douglas Allen 

Academic Editor

PLOS ONE